# The Effectiveness and Quality of Life Outcomes by Transoral Endoscopic Vestibular Thyroidectomy Using Intraoperative Indocyanin Green Fluorescence Imaging and Neuromonitoring—A Cohort Study

**DOI:** 10.3390/healthcare10050953

**Published:** 2022-05-21

**Authors:** Fadi Alnehlaoui, Mohammad Nazih Alsarraj, Zuheir Malaki, Salman Yousuf Guraya

**Affiliations:** 1Surgery Department and Oncology Services, Zulekha Hospital Sharjah, Sharjah P.O. Box 27272, United Arab Emirates; nfadi77@hotmail.com; 2Clinical Sciences Department, College of Medicine, University of Sharjah, Sharjah P.O. Box 27272, United Arab Emirates; malsarraj@sharjah.ac.ae; 3Department of Surgery, Zulekha Hospital Sharjah, Sharjah P.O. Box 27272, United Arab Emirates; zuheirmalaki.sy@gmail.com; 4Surgery Unit, College of Medicine, University of Sharjah, Sharjah P.O. Box 27272, United Arab Emirates

**Keywords:** transoral vestibular thyroidectomy, open thyroidectomy, thyroid goiter, quality of life, postoperative scarring, postoperative pain

## Abstract

Background: Transoral endoscopic vestibular thyroidectomy (TOEVT), a variant of natural orifice transluminal endoscopic surgery, offers a scar-less thyroid to young females. However, few studies have compared the effectiveness and quality of life (QoL) outcomes of the TOEVT with open thyroidectomy (OT). This is the first study in the Middle East and North Africa region that compares the effectiveness, safety profile and QoL outcomes between TOEVT with OT. Methods: We reviewed the medical records of consecutive patients with TOETV and OT at Zulekha Hospital Sharjah and Dubai United Arab Emirates, between 1 January 2019 and 1 April 2021. The data for demographics, type of surgery, operative time, blood loss, post-operative nodule size, hospital stay and post-operative complications were analyzed. We used an SF-36 questionnaire pre- and postoperatively for the assessment of QoL in both groups. Findings: Out of a total of 41 OT and 32 TOEVT procedures, 59 patients (31 TOEVT and 28 OT) fulfilled the inclusion criteria. There were 45 women and 14 men with an average age of 41 years. The mean operating time was 126 min in TOEVT and 96 min in OT (*p* = 0.000). The mean thyroid size was 5.55 cm in TOEVT and 8.76 cm in OT (*p* = 0.000). Lastly, the mean intraoperative blood loss was 39 cc and 95.7 cc in TOEVT and OT, respectively (*p* = 0.001). There was one temporary hypocalcemia and seroma in TOEVT, four cases of temporary hypocalcemia and one with minor bleeding in OT. The post-operative QoL significantly improved in all patients. However, the QoL improved more significantly in the TOEVT group for bodily pain, vitality, role emotions and cosmetic concerns (*p* = 0.000). Conclusion: The safety profile and effectiveness of the TOEVT is comparable to the OT procedure. However, TOEVT has an additional advantage of being scarless and offers a better QoL.

## 1. Introduction

The surgical approaches for benign and malignant lesions of the thyroid lesions are far ranging. Since its conception in 1880 by Kocher, the transcervical approach has remained the gold standard surgical approach for the thyroid gland [1,2]. While the transcervical open thyroidectomy (OT) offers an adequate access and exposure of the thyroid gland, this procedure often leads to visible neck scars and poor quality of life (QoL) [3]. 

Furthermore, the escalating rise in the prevalence of thyroid problems—particularly in young females—has necessitated technological advancements in endocrine surgery for more aesthetically attractive alternative approaches [4]. This has led to the introduction of a plethora of minimally invasive thyroid surgical procedures designed to reduce surgical trauma and to prevent visible surgical scars [5]. The minimally invasive thyroidectomy via endoscopic or robotic approach has shown promise in the recent past [6,7,8]. Unfortunately, each approach is a trade-off between the degree of surgical exposure and cosmesis with either an extensive tissue dissection through a distant but out of sight scar or a small but visible scar.

Endoscopic thyroid procedures are generally performed, with or without gas insufflation, via the axilla, breast, lateral neck or the anterior chest [9]. The main concern of the endoscopic or robotic approach is the need to dissect a wide flap, which tends to lose the benefits of a minimally invasive procedure [10]. As it stands, the minimally invasive thyroid surgery involves considerable tissue dissection, though via a remote or hidden incision. Henceforth, such procedures often carry a false impression about the extent of the dissection performed underneath the skin. 

From another perspective, in 2004, Kalloo et al. performed the first ground-breaking transgastric peritoneoscopy via a natural orifice transluminal endoscopic surgery (NOTES) [11]. Since then, NOTES has been clinically tested in a range of surgical disciplines including thyroid surgery using a sublingual approach for a scar-less outcome [12]. However, early reports about the effectiveness and safety profile of the NOTES thyroidectomy via the sublingual route have shown poor surgical outcomes as the procedure involves penetration of the floor of the mouth with considerable soft tissue damage and high conversion rates [5]. The NOTES approach for thyroid surgery is still considered to be technically difficult due to a limited working space and a long training curve.

The transoral endoscopic vestibular thyroidectomy (TOEVT) has emerged as a novel and attractive alternative for a scar-less thyroid surgery. This technique uses laparoscopic instruments via the oral vestibule [13]. A body of literature has shown that TOEVT is safer and esthetically more pleasing than OT [14,15]. The quality of life following transoral thyroidectomy, as estimated by neck appearance, stress, vitality and social interactions, is perceived to be better than the conventional open thyroidectomy (OT) [16,17]. 

Unfortunately, only few studies have compared the safety profile, effectiveness, complications and particularly the QoL of the patients before and after OT and TOEVT. Therefore, this study was designed to compare the safety profile, effectiveness and QoL particularly the cosmetic, functional and psychological outcomes among patients with OT and TOEVT. To our knowledge, this cohort study is the first of its kind in the Middle East and North Africa (MENA) region, which will have a substantial impact on the forthcoming technological innovations in the thyroid surgery.

## 2. Methods

### 2.1. Study Population and Setting

In this observational cohort study, we recruited consecutive patients who had thyroid surgical procedures between 1 January 2019 and 1 June 2021 at the Surgical Department of Zulekha Hospitals of Sharjah and Dubai in United Arab Emirates (UAE). Zulekha Hospital Sharjah is a 161-bedded acute-care teaching hospital, affiliated with the College of Medicine, University of Sharjah UAE. The hospital is equipped with all modern diagnostic and therapeutic facilities and caters to a wide spectrum of patients with acute and chronic illnesses from all specialties of the medical field. 

All patients were admitted through the surgical clinic of Zulekha Hospital for benign and malignant thyroid lesions. The patients who needed thyroid surgery were non randomly assigned to OT and TOEVT. After a thorough description of both procedures along with their potential benefits and risks, the patients chose the group they wanted to be in. 

The patients with advanced metastatic thyroid malignancy, previous neck surgery, substernal goiter and with previous extensive neck dissection were excluded from this study. An informed consent was taken from all patients before surgery. Pre-operative thyroid functions tests, neck ultrasound, serum calcium and indirect laryngoscopy were performed in all patients. In patients with malignant lesions of the thyroid, a CT scan of the neck and chest was performed along with and a multi-disciplinary management approach.

### 2.2. Surgical Technique of the Transoral Vestibular Thyroidectomy

The surgical steps of the TOEVT were similar to the procedure described by AJWjos [18]. Under general anesthesia with orotracheal intubation, the patient was placed in a supine and Trendelenburg’s position with neck extension. We used a long needle for the subcutaneous infiltration of the neck tissues using a 1:50,000 dilution of adrenaline prepared in 500 mL of normal saline. This infiltration helps in the creation of an anatomical plane and in the reduction of intraoperative bleeding. All steps of TOEVT are illustrated in Figure 1. 

After marking the sites for port placements, a bladeless 10 mm laparoscopic port was inserted in the midline at the vestibular region of the lower lip. Later, two short 5 mm pyramid tip ports were inserted in the lower lip at the vestibular region at the junction between the canine and the first premolar teeth on each side. Such placement avoids damage to the mental nerve. We used a 10 mm 30° view camera with an indocyanin green (ICG) infra-red scope in all TOEVT procedures. An intra-cervical pressure was maintained at 7–8 mm Hg using a continuous high flow CO_2_ insufflator. A harmonic scalpel was used for the dissection, cutting and cauterization of tissues. 

Using the harmonic scalpel, an anatomical plane was dissected up to the sternal notch inferiorly and up to the lateral borders of the sternocleidomastoid muscles on each side. The strap muscles were divided in the midline and were then retracted laterally via a transcutaneous 2/0 silk suture. The thyroid isthmus was also incised in the midline. The superior thyroid vessels were dissected and divided between clips. Then, the thyroid lobe was dissected with the identification and preservation of the recurrent laryngeal nerve (RLN). 

We routinely used specially designed long probe nerve stimulators for the identification of the RLN and the superior laryngeal nerve. When used directly over the nerve, one milliampere (mA) current was used, while a stronger current of up to 5 mA was used to identify the nerve that was embraced inside tissues. Additionally, we used the ICG fluorescence imaging to identify and confirm the location of the superior and inferior parathyroid glands (Figure 2). One ampoule of ICG was diluted in 5 mL of normal saline and a 2 mL injection of this solution was administered intravenously during the procedure. 

The parathyroid glands would be immediately visualized using the ICG infra-red arm of the telescope as shown in Figure 2. The routine applications of a nerve stimulator and intraoperative ICG fluorescence imaging, when needed, significantly enhanced the safety profile of TOEVT in our series. For a total thyroidectomy, the other lobe was similarly dissected. Lastly, the dissected thyroid lobe was removed via a 10 mm port. In the case of a total thyroidectomy, each lobe was removed separately. 

After securing the hemostasis, a hemostatic agent made of an oxidized cellulose polymer was sprinkled over the surgical field, and the strap muscles were approximated in the midline. Finally, the vestibular incisions in the oral cavity were sutured. A gravity drain was used in large goiters with difficult dissection to avoid hematoma formation. A pressure dressing in the form of a face band covering the submandibular area and the upper part of the neck was applied for 48 h.

### 2.3. Surgical Technique of the Standard Open Thyroidectomy

For the standard open thyroidectomy, we performed all procedures under general anesthesia with the patient placed in a supine and Trendelenburg’s position with neck extension. A 5 cm transverse collar incision, two fingerbreadths above the sternal notch, was made. Using a harmonic scalpel, a subplatysmal flap was created, which was dissected superiorly to the thyroid cartilage and inferiorly to the sternal notch. The strap muscles were divided in the midline, the superior thyroid vessels were ligated, and the thyroid lobe was dissected with the preservation of the RLN. The nerve stimulator was routinely used in all cases. 

As with TOEVT, the ICG fluorescence imaging was used when the parathyroid glands were not clearly visualized (Figure 2). The inferior thyroid vessels and the middle thyroid vein were ligated and divided. Finally, the thyroid lobe and isthmus were removed, and the surgical wound was closed in layers with absorbable suture. The skin was closed with subcuticular absorbable sutures. A drain was used only in cases with difficult dissection or where hematoma formation was anticipated.

In both procedures, before closing the wound, a post thyroidectomy nerve stimulation report was generated to document the viability of the nerves.

### 2.4. Postoperative Care

The patients in both groups (OT and TOEVT) were routinely administered two tablets of mefenamic acid 500 mg three times a day for mild to moderate pain and intravenous meperidine 30 mg for severe pain for two days. The patients in the TOEVT group were given prophylactic intravenous cephalosporin an hour before the procedure, and then the same antibiotic was continued intravenously for next two days. The patients were switched to oral antibiotics for the next five days. For the OT group, a single dose of prophylactic intravenous cephalosporin was given an hour before the procedure. All patients were given a liquid diet six hours after surgery. In case of an uneventful recovery, patients were discharged on the third post-operative day with a plan for a follow-up visit after a week.

We collected demographic data of patients in both groups. Additionally, we recorded data about the pre-operative histologic nature of thyroid lesion, uni- or bi-lobar, thyroid nodule or lobe size by ultrasound, type of surgery, operative time, blood loss, post-operative nodule size by histologic analysis and specimen weight, presence or absence of the parathyroid tissue in the specimen, the length of hospital stay in days and post-operative complications. A transient and permanent hypoparathyroidism was diagnosed with a serum parathyroid hormone level less than 13 pg/mL 24 h postoperatively and by 6 months, respectively. 

Mental nerve injury was characterized as lower lip paresthesia after surgery. We defined postoperative hematoma as an obvious collection of seroma and blood that led to a neck swelling and required a drainage procedure.

For a more objective analysis of the surgical outcome and the QoL of patients in both groups, we used the validated SF-36 questionnaire [19] pre-operatively as well as during sixth week post-operatively. All patients were informed about the nature and purpose of the survey, and we obtained informed consent from all patients. The SF-36 questionnaire contains eight domains of physical functioning, role-physical, bodily pain, general health, vitality, social functioning, role-emotional and mental health. Each domain consists of a five-point Likert scale from one to five, higher the score better the outcome.

### 2.5. Statistical Analysis

We used SPSS version 20.0 for the quantitative analysis of the data. The descriptive analysis was done using frequency distributions, mean and standard deviation and a range of other factors used in the study. The selected items from both groups were compared using the Fisher exact test for categorical variables, while for the continuous variables, an independent two-tailed paired *t*-test was applied. The mean (standard deviation) and number (percentages) were expressed as continuous and categorical variables. Finally, for the comparison of pre-post QoL, we used a dependent *t*-test. A value of *p* < 0.05 was considered statistically significant.

## 3. Results

A total of 41 OT and 32 TOEVT were performed during the study period. After excluding cases with advanced thyroid malignancy needing neck dissection or palliative care, a complete data of 59 patients, 31 TOEVT and 28 OT, was retrieved. There were 45 women and 14 men with an average age of 41 years (age range 23 to 48 years) and an average BMI of 30. Other demographic details of all patients are shown in Table 1.

A comparative analysis of surgical details between TOEVT and OT is outlined in Table 2.

The mean operating time in the TOEVT group was 126.37 min, while the same was recorded to be 96.39 min in the OT group (*p =* 0.000). The average thyroid size was 5.55 cm in TOEVT and 8.76 cm in the OT group (*p =* 0.000). There was insignificant difference in the average size of the thyroid nodule between both groups. Drains were used in six TOEVT and 20 OT patients (*p =* 0.000). There were minor complications in both groups that did not affect the surgical outcomes in our series.

Table 3 presents the range of histological diagnoses from the resected thyroid specimen in our series.

Table 4 provides a comparative analysis of the pre-post-operative QoL of patients in the TOEVT and OT groups. We found that the QoL significantly improved post-operatively in all eight domains in both groups. However, the QoL improved more significantly in TOEVT than in the OT group as the TOEVT patients had significantly higher QoL scores in bodily pain, vitality, role emotions and cosmetic domains (*p =* 0.000).

To eliminate the sample selection bias, we performed propensity score matching using the Nearest Neighbor Matching technique under logit regression model of dependent variable (Surgery type TOEVT vs. OT dummy variable) and independent variables (gender, age, BMI, blood loss and operating time) using the STATA 15 software. This model yielded a matched sample of 21 patients in each group. The corresponding outcomes of the propensity score matching were similar to the results of the standard statistical analyses performed in our study as shown in Table 5, Table 6 and Table 7.

The peri-operative profiles of a patient including her pre-operative and post-operative views at first and second weeks are shown in Figure 3.

## 4. Discussion

Our study demonstrates the efficacy and safety profile of the TOEVT compared with the OT. In our study, there was no conversion of TOEVT to OT. The demographic profile of patients, the range of thyroid conditions, mean hospital stay and rate of complications were similar in both groups. However, the size and weight of thyroid gland were markedly smaller in the TOEVT compared with in the OT group. 

Another difference was less mean intraoperative blood loss in the TOEVT than in the OT approach, though not statistically significant. This is essentially attributed to more meticulous attention to hemostasis during TOEVT as the laparoscopic image is magnified, and the surgeon has a better control on the surgical field. Lastly, very few patients had drains in TOEVT than in the OT group. Once again, a superior hemostasis and less intraoperative blood loss obviated the need for a drain that can potentially lead to post-operative morbidity.

We did not find a significant difference in the mean age of patients between TOEVT and OT groups. In contrast, in the study by Kasemiri et al., comparing the QoL between TOEVT and OT, the investigators found that the TOEVT group had younger mean age with smaller thyroid nodules [20]. From the TOEVT group, the study has also reported a mean nodule size of 2.8 cm, which matches with the mean nodule size of 2.9 cm in our TOEVT group. 

We found a statistically smaller mean thyroid size of 5.55 cm in TOEVT compared with 8.76 cm in OT in our series (*p =* 0.000). These findings emphasize the need for the right selection of cases with small thyroid gland and nodule size for the TOEVT approach. Researchers have argued that division of the thyroid gland and its extraction via a small vestibular incision can potentially distort the thyroid capsule that can undermine the accuracy of histological diagnosis of follicular neoplasms [21]. 

To avoid such an unwanted outcome, a new technique of transoral submental thyroidectomy (TOaST) has been introduced, which offers a slightly longer submental incision for the extraction of larger thyroid glands [17,21]. Though this approach is not completely scarless, thyroid glands larger than 5 cm can be extracted via the submental incision without endangering its capsular integrity [22]. In our series, the mean operating time was 126 min for TOEVT and 96 min for OT cases. 

In the study by Anuwong et al., the authors reported a mean operating time of 100.8 min for TOEVT and 79.4 min for OT [23]. Another report demonstrated a mean operative time of 87.6 min for lobectomy and 107.6 min for a total thyroidectomy [24]. Other studies have also reported a longer mean operating time by endoscopic thyroid surgery [25,26]. A high volume of patients, greater surgical skills and experience can potentially reduce the time needed for endoscopic thyroid surgery. A slightly longer operating time for TOEVT in our series is partly attributed to the recent introduction of TOEVT as a new innovative surgical technique in our hospital that takes time in creating a harmonized surgical teamwork [27].

In our study, the mean intraoperative blood loss was 39 cc in TOEVT and 95.7 cc in OT cases (*p* = 0.001). Other studies have also reported more intraoperative blood loss in OT than in TOEVT [28,29]. Furthermore, in our study, drains were placed in only 6/27 TOEVT cases, while drains were used in 20/24 OT cases. These findings reaffirm a superior hemostatic control of TOEVT due to a higher magnification of telescope and the use of energy devices, such as ligature or harmonic scalpel. The use of conventional energy devices for electrocautery are generally attended by long-term complications [30], which can be avoided by using modern devices with less collateral damage and with more local control on cutting and coagulation [30].

It is pertinent to understand that endoscopic procedures have a limited value for a locally advanced or metastatic thyroid malignancy as these conditions require extensive neck dissection. Such advanced oncological procedures cannot be performed for endoscopic thyroid approaches. By and large, the core principles of surgical oncology demand a wide exposure of cancer including its adjacent structures for enbloc dissection and removal of the tumor along with its drainage lymph nodes [31,32]. Nevertheless, as evident from our cohort, all types of thyroid surgical procedures can be effectively performed by the TOEVT for benign and small-sized malignant tumors of the thyroid gland.

In both the TOEVT and OT groups, we observed minor complications that did not affect the surgical outcomes in our series. We routinely used intraoperative ICG fluorescence imaging for the identification and perfusion status of the parathyroid glands. Post thyroidectomy, transient and permanent hypocalcemia have been reported in 15 to 30% and 1 to 3% of patients, respectively [33]. We had a very small number of patients with temporary hypocalcemia in both groups who rapidly recovered without adverse outcome. Another reason for the very low rate of complications in our study was the routine use of intraoperative neuromonitoring for the superior laryngeal and RLN. In our study, there was no laryngeal nerve injury in both groups.

The reported rates of transient, permanent and bilateral RLN palsy in transoral thyroidectomy are 1.86%, 0.1% and 0%, respectively [34]. These extremely low rates of RLN injury in the TOEVT result from the endoscopic magnification of the operative field, intraoperative neuromonitoring, top-down view and a cranio-caudal dissection [35]. Anuwong proposed that the central incision should be made at the mid-oral vestibule, while the two 5 mm ports incisions should be limited to the canines as the branches of the mental nerves emanate from the mental foramen at the first premolar teeth [18]. 

Thermal damage to the nerve should be considered while using ultrasound devices. Lastly, a thyroid specimen larger than 4 cm is difficult to extract from the vestibular incision, and most thyroid specimens need to be fragmented into smaller parts before their removal. TOEVT is a potentially type II clean-contaminated wound and, in our series, we continued a broad-spectrum antibiotic for seven days postoperatively. This prophylactic antibiotic regime dramatically kept the risk of surgical site infection to the bare minimum.

We found that the QoL significantly improved post-operatively in all eight components in both TOEVT and OT groups. However, the QoL improved more substantially in the TOEVT than in the OT group. Bodily pain, vitality, role emotion and cosmetic concerns obtained the highest score in the TOEVT group (*p =* 0.000). These findings prove a superior QoL of TOEVT over OT. 

Disturbed swallowing and a limitation of neck movements due to fibrosis of the cervical corridor, damage to the branches of the mental nerve with associated chin numbness and a pulling sensation in the neck due to the subplatysmal dissection have been reported to adversely affect the QoL following TOEVT [36]. No such complications were reported in our study. In surgical patients, vitality, social and mental health are ingredients for a rapid recovery after surgery. However, our study demonstrates findings from a short-term follow up from a single center.

## 5. Study Limitations

Our study presents a small cohort of patients with a short-term follow up. Findings of our cohort may not be strong enough to draw major conclusions. Large multi-center clinical studies are required to validate the safety profile and effectiveness of TOEVT. In addition, being a newly emerging surgical technique, less extensive surgical procedures were performed in TOEVT. With the growing experience and confidence of the surgical team, the scope of its surgical indications will certainly expand.

## 6. Conclusions

Based on the findings of our single center study, the TOETV was found to be a safe, effective and feasible surgical approach for a range of thyroid disorders. The TOEVT offers scarless surgery with superior cosmetic and other QoL outcomes compared with in the OT patients. The TOETV is associated with a longer operating time, less blood loss and a similar rate of complications compared with OT. Routine laparoscopic surgery instruments can be effectively used for TOETV.

## Figures and Tables

**Figure 1 healthcare-10-00953-f001:**
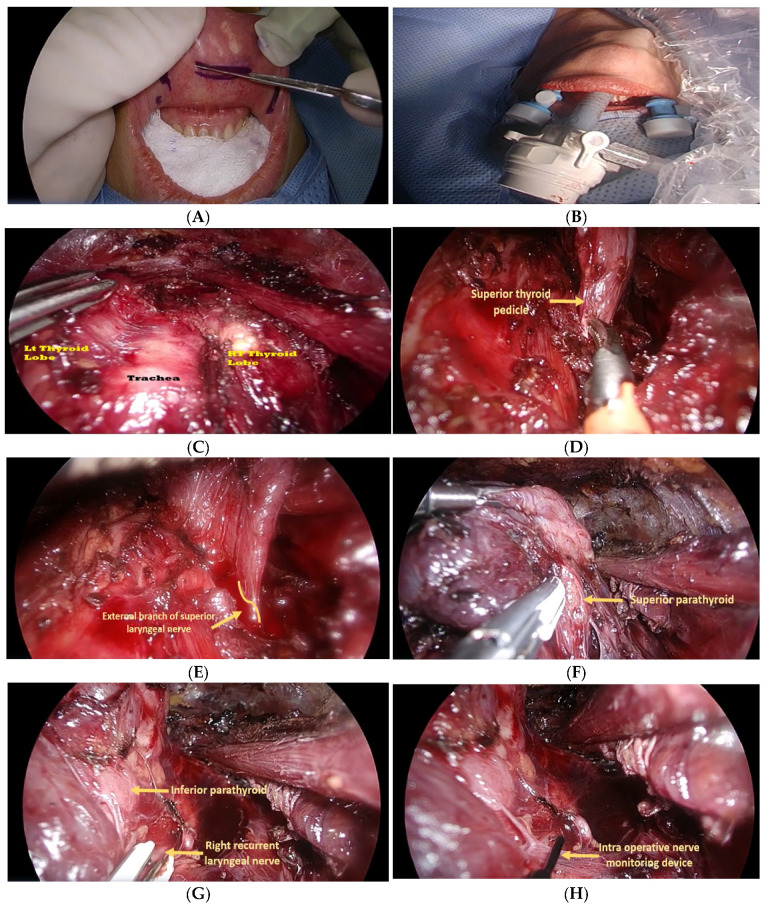
Surgical steps during the transoral vestibular thyroidectomy (**A**), marking for incisions and trocar placements (**B**), placement of a midline 10 mm and two 5 mm ports (**C**), endoscopic view of the thyroid lobes and trachea. Identification of the superior thyroid pedicle (**D**), the external branch of the superior laryngeal nerve (**E**), the superior parathyroid gland (**F**), the inferior parathyroid gland and the recurrent laryngeal nerve (**G**) and the use of intraoperative nerve stimulator (**H**).

**Figure 2 healthcare-10-00953-f002:**
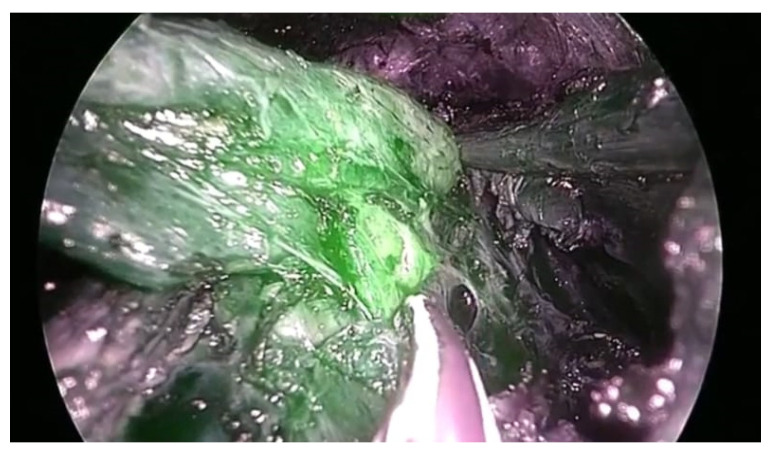
The use of indocyanin green fluorescence imaging for the intraoperative identification of the parathyroid glands during the transoral vestibular thyroidectomy.

**Figure 3 healthcare-10-00953-f003:**
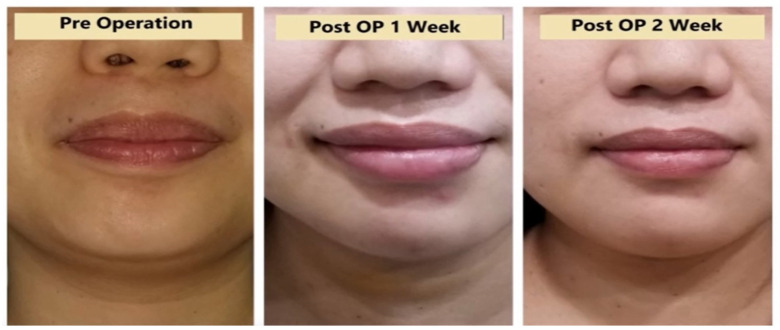
Profiles of a patient at the pre-operative stage and at first and second weeks following transoral endoscopic vestibular thyroidectomy.

**Table 1 healthcare-10-00953-t001:** Demographic characteristics of patients in transoral endoscopic vestibular and open thyroidectomy groups (*n* = 59).

Factor	TOEVT * (*n* = 31)	OT ^#^ (*n* = 28)	*p*-Value
Gender	No (%)	No (%)	
Female	26 (83.8)	19 (67.8)	0.064
Male	5 (16.1)	9 (32.1)	
Age, mean (SD) in years	42.37 (9.33)	44.65 (10.68)	0.057
BMI, mean (SD)	28.20 (4.99)	31.23 (7.66)	0.017

Note: * denotes transoral endoscopic vestibular thyroidectomy and ^#^ open thyroidectomy.

**Table 2 healthcare-10-00953-t002:** A comparison of the surgical outcomes between transoral endoscopic vestibular and open thyroidectomy (*n* = 59).

Factor	TOEVT * (*n* = 31)	OT ^#^ (*n* = 28)	*p*-Value
Type of Thyroid Surgery	No (%)	No (%)	
Lobectomy	20 (64.5)	9 (32.1)	0.000 *
Total Thyroidectomy	11 (35.4)	19 (67.8)	
Operation time, mean (SD), minutes	126.37 (61.31)	96.39 (26.05)	0.079
Blood loss, mean (SD), cc	39.00 (60.29)	95.71 (69.17)	0.001 *
Thyroid size, mean (SD), cm	5.55 (1.94)	8.76 (2.23)	0.000 *
Nodule size, mean (SD), cm	2.96 (1.72)	3.10 (1.89)	0.081
Thyroid weight, mean (SD), gram	15.88 (1.64)	56.18 (4.06)	0.000 *
**Drain**			
Yes	6 (19.3)	20 (71.4)	0.000 *
No	25 (80.6)	8 (28.5)	
**Complications**			
None	29 (93.1)	23 (82.1)	0.069
Temporary hypocalcemia	1 (3.2)	4 (14.2)	
Minor Bleeding	0 (0.0)	1 (3.5)	
Seroma	1 (3.2)	0 (0.0)	

Note: * denotes transoral endoscopic vestibular thyroidectomy and ^#^ open thyroidectomy.

**Table 3 healthcare-10-00953-t003:** Histological diagnoses of the thyroid glands in the transoral endoscopic vestibular and open thyroidectomy procedures (*n* = 59).

Histological Diagnosis	TOEVT * (*n* = 31)	OT ^#^ (*n* = 28)
Follicular adenoma	14	4
Multinodular goiter	9	17
Papillary carcinoma	6	5
Hurthle’s cell adenoma	2	0
Grave’s disease	0	2

Note: * denotes transoral endoscopic vestibular thyroidectomy and ^#^ open thyroidectomy.

**Table 4 healthcare-10-00953-t004:** A comparison of the quality of life pre-post transoral endoscopic vestibular and open thyroidectomy (*n* = 59).

Factor	Overall			TOEVT *			OT ^#^		
	Pre	Post	*p*-Value	Pre	Post	*p*-Value	Pre	Post	*p*-Value
Physical function	3.3	4.1	0.01	3.2	4.1	0.03	3.3	4.1	0.013
Bodily pain	3.9	4.6	0.00	3.1	4.8	0.00 *	3.7	4.4	0.001
General Health	3.6	4.3	0.00	3.7	4.5	0.04	3.4	4.2	0.002
Vitality	3.3	4.1	0.01	3.0	4.6	0.00 *	3.2	3.8	0.012
Social function	3.6	4.1	0.04	3.9	4.3	0.08	3.2	3.8	0.026
Mental health	3.3	4.1	0.01	3.2	4.1	0.05	3.3	4.1	0.014
Role emotion	3.3	4.3	0.00	3.0	4.7	0.00 *	3.2	4.1	0.001
Cosmetic concerns	3.2	4.2	0.00	3.1	4.3	0.00 *	3.3	4.3	0.011

Note: * denotes transoral endoscopic vestibular thyroidectomy and ^#^ open thyroidectomy.

**Table 5 healthcare-10-00953-t005:** A comparison of the surgical outcomes between transoral endoscopic vestibular and open thyroidectomy after propensity score matching (*n* = 34).

Factor	TOEVT * (*n* = 17)	OT ^#^ (*n* = 17)	*p*-Value
Type of Thyroid Surgery	No (%)	No (%)	
Lobectomy	11 (65)	8 (47)	0.128
Total Thyroidectomy	6 (35)	9 (53)	
Operation time, mean (SD), minutes	101.36 (32.74)	77.51 (22.04)	0.092
Blood loss, mean (SD), cc	39.00 (19.29)	99.18 (73.42)	0.023 *
Thyroid size, mean (SD), cm	5.11 (1.97)	5.27 (2.64)	0.920
Nodule size, mean (SD), cm	2.90 (1.38)	3.12 (1.40)	0.776
Thyroid weight, mean (SD), gram	13.09 (1.57)	16.42 (2.94)	0.137
**Drain**			
Yes	3 (18)	12 (71)	0.004 *
No	14 (82)	5 (29)	
**Complications**			
None	17 (100)	14 (82)	0.413
Temporary hypocalcemia	0 (0.0)	2 (12)	
Minor Bleeding	0 (0.0)	1 (6)	
Seroma	0 (0.0)	0 (0.0)	

Note: * denotes transoral endoscopic vestibular thyroidectomy and ^#^ open thyroidectomy.

**Table 6 healthcare-10-00953-t006:** Histological diagnoses of the thyroid glands in the transoral endoscopic vestibular and open thyroidectomy procedures after Propensity Score Matching (*n* = 34).

Histological Diagnosis	TOEVT * (*n* = 17)	OT ^#^ (*n* = 17)
Follicular adenoma	8	3
Multinodular goiter	5	10
Papillary carcinoma	3	4
Hurthle’s cell adenoma	1	0

Note: * denotes transoral endoscopic vestibular thyroidectomy and ^#^ open thyroidectomy.

**Table 7 healthcare-10-00953-t007:** A comparison of the quality of life pre-post transoral endoscopic vestibular and open thyroidectomy after propensity score matching (*n* = 34).

Factor	Overall			TOEVT *			OT ^#^		
	Pre	Post	*p*-Value	Pre	Post	*p*-Value	Pre	Post	*p*-Value
Physical function	3.5	4.3	0.015	3.3	4.1	0.132	3.7	4.6	0.063
Bodily pain	3.1	3.5	0.021	4.1	4.8	0.045	3.0	3.9	0.043
General Health	3.8	4.7	0.003	3.6	4.5	0.037	3.2	4.3	0.032
Vitality	3.5	4.4	0.008	3.4	4.3	0.048	3.5	4.3	0.016
Social function	3.8	4.3	0.047	3.9	4.3	0.264	3.8	4.4	0.082
Mental health	3.3	4.3	0.010	3.2	4.0	0.145	3.5	4.7	0.035
Role emotion	3.4	4.5	0.001	3.2	4.4	0.010	3.6	4.5	0.029
Cosmetic concerns	3.3	4.5	0.004	3.2	4.3	0.012	3.4	4.7	0.007

Note: * denotes transoral endoscopic vestibular thyroidectomy and ^#^ open thyroidectomy.

## Data Availability

Data are available upon reasonable request. All relevant data are provided in the manuscript. Sharing of the original data is not permitted within the achieved permissions.

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
