# Peer review of "The Effectiveness and Quality of Life Outcomes by Transoral Endoscopic Vestibular Thyroidectomy Using Intraoperative Indocyanin Green Fluorescence Imaging and Neuromonitoring—A Cohort Study"

_healthcare, 2022, doi:10.3390/healthcare10050953_

Round 1

Reviewer 1 Report

Line 33 - I suggest changing "bodily pain" to "pain";

Line 45 - I suggest changing "ailments" to "disturbs";

The image ofter the line 139 seems too stretched;

I noticed that 11 papillary thyroid carcinomas were diagnosed. The TOEVT technique did not imply difficulties in the histological evaluation of these patients, for example in the description of capsular invasion or surgical margins? Were all neoplasms intraglandular?

You performed 20 lobectomies (TOEVT), did any of these require a total thyroidectomy? Did you document any complications or difficulties in reoperations?

Author Response

Line 33 - I suggest changing "bodily pain" to "pain";

Thanks for the suggestion. However, we used the validated SF-36 questionnaire to determine the quality of life of patients in TOEVT and and OT. This validated SF-36 questionnaire has specific variables including 'bodily pain' and this can not be changes as, by doing so, the harmony of terminology will be affected. 

Line 45 - I suggest changing "ailments" to "disturbs";

Thanks. Corrected

The image ofter the line 139 seems too stretched;

The image is readjusted. However, during the final proof reading and copy editing, this image will be edited again as the figure is genuine and contains a high quality pixelated image. 

I noticed that 11 papillary thyroid carcinomas were diagnosed. The TOEVT technique did not imply difficulties in the histological evaluation of these patients, for example in the description of capsular invasion or surgical margins? Were all neoplasms intraglandular?

Thanks for the observation. We did not face any difficulty during the histological evaluation of papillary carcinomas as the glands were divided inside the neck and taken out in two pieces without any difficulty. 

Yes, all neoplasms were intra glandular. 

You performed 20 lobectomies (TOEVT), did any of these require a total thyroidectomy? Did you document any complications or difficulties in reoperations?

No. There is no reoperation of the 20 lobectomies performed by TOEVT so far. 

Reviewer 2 Report

The authors compared transoral endoscopic vestibular thyroidectomy and open thyroidectomy and the content of the article was interesting.

However, I think some additional information would be needed for publication.

  1. In methods section, the authors described "The patients who needed thyroid surgery were randomly assigned OT and TOEVT". Did this mean that this study was  randomized study?. However, the background of patients ware quite different, in regard to BMI, type of thyroid surgery, thyroid size, drain, historical diagnosis, etc.
  2. In surgery for papillary carcinoma, did you do lymph node dissection in central lesion?  
  3. The authors described QOL was better in the patients with TOEVT. However, the background of the patients was different, in that the proportion of total thyroidectomy and drain was low in the TOEVT patients compared with OT. I think these factors could affect QOL. So I think propensity score matching method etc. should be needed.

Author Response

The authors compared transoral endoscopic vestibular thyroidectomy and open thyroidectomy and the content of the article was interesting.

However, I think some additional information would be needed for publication.

  1. In methods section, the authors described "The patients who needed thyroid surgery were randomly assigned OT and TOEVT". Did this mean that this study was  randomized study?. However, the background of patients ware quite different, in regard to BMI, type of thyroid surgery, thyroid size, drain, historical diagnosis, etc.
  2. Thanks indeed for pointing out this error. We have modified the text as follows; The patients who needed thyroid surgery were non randomly assigned to OT and TOEVT. After a thorough description of both procedures along with their potential benefits and risks, the patients chose the group they wanted to be in.  
  3. In surgery for papillary carcinoma, did you do lymph node dissection in central lesion?  
  4. No lymph node dissection was done in surgery for papillary carcinoma in our series. 
  5. The authors described QOL was better in the patients with TOEVT. However, the background of the patients was different, in that the proportion of total thyroidectomy and drain was low in the TOEVT patients compared with OT. I think these factors could affect QOL. So I think propensity score matching method etc. should be needed.
  6. We appreciate your observation and, consequently, we have performed the propensity score matching of the series. The output tables and a detailed description is attached for your kind review.

Reviewer 3 Report

The manuscript "The Effectiveness and Quality of life Outcomes by Transoral Endoscopic Vestibular Thyroidectomy Using Intraoperative Indocyanin Green Fluorescence Imaging and Neuromonitoring- A Cohort Study" aims to compare TOEVT and OT regarding on effectiveness, safety and quality of life of patients. The authors demonstrate although safety and effectiveness of TOEVT was similar to OT, the QoL improved in TOEVT. The manuscript is interesting and well written.

The procedure- TOEVT is not new  among the medical and scientific community but the manuscript demonstrates the advantages of TOEVT comparing with OT, using intraoperative indocyanin greeen fluorescence imaging and neuromonitoring.  The main concern is the selection of patients to TOEVT procedure. Can the authors provide some information on this?

Author Response

The manuscript "The Effectiveness and Quality of life Outcomes by Transoral Endoscopic Vestibular Thyroidectomy Using Intraoperative Indocyanin Green Fluorescence Imaging and Neuromonitoring- A Cohort Study" aims to compare TOEVT and OT regarding on effectiveness, safety and quality of life of patients. The authors demonstrate although safety and effectiveness of TOEVT was similar to OT, the QoL improved in TOEVT. The manuscript is interesting and well written.

The procedure- TOEVT is not new  among the medical and scientific community but the manuscript demonstrates the advantages of TOEVT comparing with OT, using intraoperative indocyanin greeen fluorescence imaging and neuromonitoring.  The main concern is the selection of patients to TOEVT procedure. Can the authors provide some information on this?

Thanks for the comment. A response is added int the manuscript as follows. The patients who needed thyroid surgery were non randomly assigned to OT and TOEVT. After a thorough description of both procedures along with their potential benefits and risks, the patients chose the group they wanted to be in. 

Round 2

Reviewer 2 Report

The authors responded to the points that I recommended.

I think the extent of thyroidectomy, thyroid size, and thyroid weight would affect patients QOL. However, after propensity score matching, the proportion of total thyroidectomy was significantly different. I think the proportion of the these factors should be identical after propensity score matching when QOL is compared between 2 groups.

Author Response

I think the extent of thyroidectomy, thyroid size, and thyroid weight would affect patients QOL. However, after propensity score matching, the proportion of total thyroidectomy was significantly different. I think the proportion of the these factors should be identical after propensity score matching when QOL is compared between 2 groups

Thanks for the second round of review. Following the remarks, we have run the stats for PSM and have added the following in the manuscript;

 For eliminating the sample selection bias, we performed the propensity score matching using the Nearest Neighbor Matching technique under logit regression model of dependent variable (Surgery type TOEVT vs OT dummy variable) and independent variables (gender, age, BMI, blood loss, thyroid size, nodule size, thyroid weight, and operating time) using the STATA 15 software. This model yielded a matched sample of 17 patients in each group. The corresponding outcomes of the propensity score matching were similar to the results of the standard statistical analyses performed in our study as shown in Tables 5, 6 and 7.

The revised manuscript is enclosed for your kind review.
